# First record of *Sigmodon minor* (Rodentia) in the early Blancan of central Mexico: Asymmetrical dispersal from the Great Plains and paleoecology inferences

Adolfo Pacheco-Castro[1,2], Oscar Carranza-Castañeda[3], Xiaoming Wang[1,4]*

1 Department of Vertebrate Paleontology, Natural History Museum of Los Angeles County, Los Angeles, California, United States of America, 2 Facultad de Ciencias Naturales, Universidad Autónoma de Querétaro, Querétaro, Mexico, 3 Instituto de Geociencias, Universidad Nacional Autónoma de México, Querétaro, Mexico, 4 Department of Earth Sciences, University of Southern California, Los Angeles, California, United States of America

* xwang@nhm.org

## Abstract

In this study, we report the first occurrence of *Sigmodon minor* (Cricetidae, Rodentia) in Mexico, from the early Blancan of the Pliocene in the San Miguel de Allende Basin, Guanajuato. This record represents an early and rapid dispersal of these rodents from the Great Plains to lower latitudes, likely driven by climatic fluctuations during the late Neogene that favored the expansion of grassland biomes. The fossils described here are among the most complete for this species, including well-preserved mandibles and maxillary elements, found in association with megafauna and supported by radiometric age data. Detailed comparisons of molar dental occlusal structures and evolutionary stages with other contemporary records from North America were conducted. Palaeoecological inferences based on body size suggest a predominance of open grassland ecosystems interspersed with wetland niches, reflecting a complex mosaic of environmental conditions.

## Introduction

During the Pliocene, the climatic conditions of North America were marked by declining temperatures and increased aridity. This led to the proliferation of open ecosystems and the prevalence of temperate grasslands and tropical savannas [1,2]. Under these conditions, the faunas of the Great Plains began to expand on the continent, i.e., equids, camelids, proboscideans, carnivores, and rodents adapted to these open environments became common members throughout North America [3,4]. One of these members were the species of the genus *Sigmodon* (Cricetidae, Sigmodontinae), the cotton rats, whose first record dates back to the early Blancan of the Meade basin, Kansas [5].

**Data availability statement:** All relevant data are within the paper and its Supporting information files.

**Funding:** This work was supported by National Science Foundation [EAR-1949742, EAR-1655720], and UNAM DGAPA PAPIIT IN102425. The funders had no role in study design, data collection and analysis, decision to publish, or preparation of the manuscript.

**Competing interests:** The authors have declared that no competing interests exist.

This first record of cotton rats corresponds to the appearance of the species *Sigmodon minor*, which began a rapid expansion in North America, initially distributed in the central and western regions during the early Blancan North American Land Mammal Age (NALMA) [5,6], and later during the late Blancan to the eastern region of the continent, such as in Florida [7]. However, despite the similarity in megafauna species between the Great Plains and Mexico, this species has not been documented outside of Unite States of America (USA) thus far.

It was intriguing that a widespread component of the Blancan faunas, such as *Sigmodon minor*, was absent in the southern part of North America, i.e., in Mexico. This is noteworthy, particularly in the best-studied basins, such as in the north of the country in the locality of La Concha, related to the Yepómera Basin [8], and in the central region where the San Miguel de Allende Basin (SMA) is situated [9]. Interestingly, these are faunas that have been characterized by a their high richness of sigmodontine species, among which are members of the genus *Prosigmodon*, closely related to the genus *Sigmodon* [9–11]. Furthermore, Mexico is one of the countries with the greatest modern diversity of *Sigmodon* species, and molecular studies have proposed that important evolutionary processes within the lineage took place in this region. This is the case for the evolution of the *Sigmodon hispidus* species group that diversified 4 Ma ago from a common ancestor with *Sigmodon ochrognathus* [12], which are distributed from Nebraska to Venezuela with a high density in Mexico [13].

Another relevant factor is that *Sigmodon minor* is a species used as a paleoecological indicator of more temperate and humid conditions within grassland biomes [6,14,15]. Therefore, the stratigraphic record of this species is essential for understanding the dispersal patterns of North American faunal assemblages into the more southern latitudes of North America, where a mosaic of species with distinct physical boundaries evolved in response to periods of climate variability [4].

Hence, in this study we present the first *Sigmodon minor* record in Mexico, recovered from the early Blancan deposits of the San Miguel de Allende basin, Guanajuato. We undertake a comparative study with North American *S. minor* records, discussing the possibility of its dispersal from the Great Plains during the dispersion of grass-dominated ecosystems. Additionally, we study dental morphology and body size to assess whether this discovery suggests an early arrival of this species, potentially driven by the more temperate and humid environments prevalent in central Mexico during the Pliocene.

### Geological setting

The San Miguel de Allende Basin is located in the State of Guanajuato, central Mexico, and represents the most studied Neogene paleontological record in the country (see several references in Carranza-Castañeda 2006) [16]. It is located between 20°and 21°N latitude within the limits of the geological province known as the Trans-Mexican Volcanic Belt (Fig 1). Its sediments are fluvio-lacustrine and were deposited during the Late Miocene–Early Pleistocene at the same time that a mafic episode occurred in the region that developed several associated volcanic structures [17].

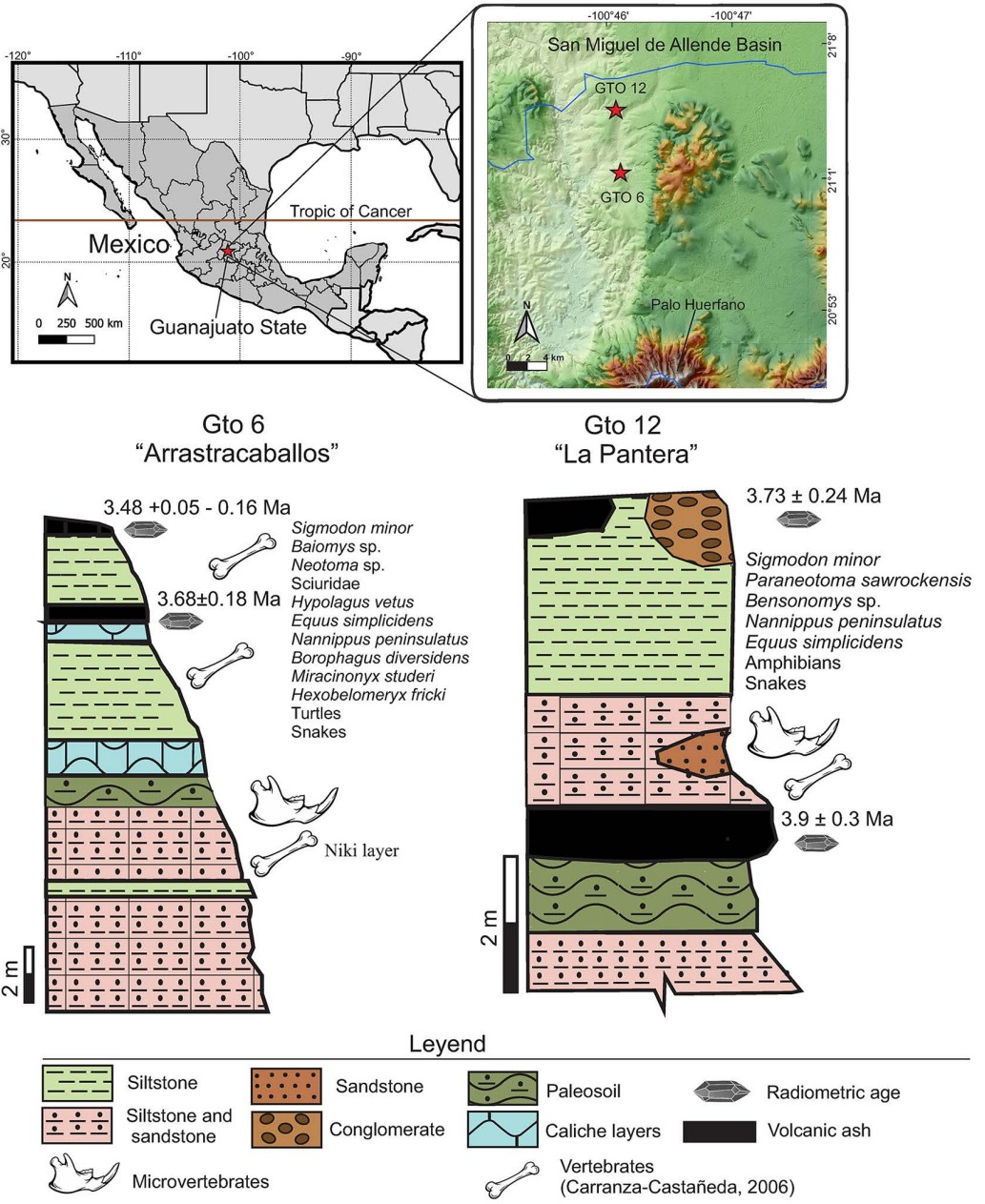

**Fig 1. Geographic and stratigraphic context of the localities in San Miguel de Allende Basin.** Location of San Miguel de Allende Basin in central Mexico and stratigraphy columns of the localities GTO 6 "Arrastracaballos" and GTO 12 "La Pantera".

Nearly a hundred fossil sites have been identified within this basin, spanning from the late Hemphillian to the Irvingtonian NALMA, with no evidence of late Blancan. Extensive research efforts have been directed towards understanding the boundary between Hemphillian and Blancan, as well as the reshaping of ecosystems during the Great American Biotic Interchange (GABI). These studies suggest the presence of an effective land corridor in central Mexico between North and South America [18,19]. Consequently, a well-studied stratigraphic relationship of various localities aids in establishing a composite stratigraphic sequence with mammal assemblages and radiometric ages [16,20].

In particular, this work describes the fossil material of Sigmodontinae rodents from three early Blancan localities: GTO 6 "Arrastracaballos," GTO 12 "La Pantera" and GTO 19 "Pecos" (Fig 1). These were discovered by Harley Garbani and Oscar Carranza-Castañeda in the 1980s-1990s, but only two, GTO6 and GTO 12, were extensively surveyed. Currently, the exact location of GTO 19 "Pecos" is unknown, as there has been an accelerated change in the rural landscape of this area. But it is known to be associated with early Blancan sediments from the Rancho Viejo Area, in particular with the localities GTO 5 "Miller Place," GTO 33 "Pork Chop" and GTO 14 "Cuesta Blanca," where *Neochoerus cordobai* and *Glossotherium* have been collected [16]. Furthermore, in GTO 19 the rodent material was associated to elements of the equid *Nannippus peninsulatus* (MPGJ 2696 and MPGJ 2697), which is considered an index fossil for this age.

Towards the south of this area is the locality GTO 6 "Arrastracaballos", at 21°00.92'N, 100°45.31'W, which corresponds to the most important locality of the early Blancan in Mexico, owing to the diversity and abundance of specimens found here. Records of *Nannippus peninsulatus*, *Borophagus diversidens*, *Miracinonyx studeri*, *Taxidea mexicana*, *Trigonictis*, *Canis* cf. *C. lepophagus*, *Hexobelomeryx fricki*, *Hemiauchenia blancoensis* and *Hypolagus vetus,* are distributed in all the sequence [16]. The sediments in this locality are alternations of siltstones and sandstones with some intermediate caliche layers. Near the base of the column is the stratum identified as the Niki Layer, in which abundant microvertebrates are found (Fig 1). In the upper part of the section, a volcanic ash is deposited, which is dated in this work and has an age of 3.68±0.18 Ma (S1 Appendix).

In a distinct sector of the basin, known as the Los Galvanes area, is the locality GTO 12 "La Pantera", situated at 21°03.78'N, 100° 46.13'W. This locality is particularly rich in fossils of small vertebrates, ranging from fish, reptiles, amphibians, and mammals, to the equids *Nannippus peninsulatus* and *Plesippus simplicidens*. The sedimentary deposits with a high density of fossils consist of normally graded siltstone and sandstone. These deposits are interbedded between two layers of volcanic ashes, each with precisely determined radiometric ages. The lower ash layer, which overlays a paleosol with concretions, dates to approximately 3.9 million years ago, based on fission track analysis of a single zircon crystal [21]. In the upper part of the sequence, covering the sediments, there is volcanic ash that was dated in this study at the Laboratorio de Estudios Isotópicos del Instituto de Geociencias, UNAM (LEI) by laser ablation in zircons, establishing its age at approximately 3.73 Ma±0.24 (S1 Appendix).

These localities are among the most significant in the region due to their high concentration of microvertebrate remains. Corrosion observed on some molar crowns suggests exposure to gastric acids, indicating that the accumulation at GTO 12 and GTO 19 likely resulted from predation by an unidentified predator. In contrast, at GTO 6, microvertebrate remains occur in stratigraphic concordance and in association with megafauna, suggesting accumulation through natural hydraulic sorting within an ancient fluvial system.

## Materials and methods

### Collecting methods and specimen repository

Rodent fossils were collected mainly through screen-washing techniques, although the most complete specimens were surface-collected from each site. Fossils were prepared and cataloged in the Colección del Laboratorio de Paleontología of Instituto de Geociencias, Universidad Nacional Autónoma de México (MPGJ). All necessary permits were obtained for the described study, which complied with all relevant regulations.

### Dental terminology and measurements

The notations for the dental position are I1, M1, M2, and M3 for the upper teeth and i1, m1, m2, and m3 for the lower teeth. Dental terminology was based on Reig et al and Martin et al [22,23]. However, for molars M1 and m1 the terms anterocone and anteroconid are used instead of the procingulum. Although it is known that in sigmodontines, the anterior region of the molar is composed of a system of minor cusps [24], it is simpler in *Sigmodon* species, where m1 and M1

have a procingulum as a single cusp that occupies an important diagnostic area in the anterior region of the molar, formed by the fusion between the conules (ids) without cingulate or accessory cusps (S1 Fig).

The Images of the specimens in collections were taken with a Keyence VHX 4000 in the Entomology Department of the Natural History Museum of Los Angeles County, and Leica camera coupled to a stereomicroscope in the Instituto de Geociencias, Universidad Nacional Autónoma de México. Anteroposterior length and width measurements of the specimens were obtained using ImageJ software The anteroposterior length and width measurements of the specimens were taken using the ImageJ software [25].

## Paleobiogeographic models and body mass estimation

We utilized the Paleobiology Database (Paleobiodb.org) for biogeographic correlations in North America and supplemented this information with the literature to establish precise temporal boundaries for *Sigmodon minor* records. The database, as of January 6, 2025, includes 78 records of *Sigmodon minor*, encompassing both the *S. minor/minor* and *S. minor/medius* varieties (see S1 Table). The Blancan age records in the database were meticulously reviewed through an updated literature analysis to ensure accuracy, with the aim of proposing a biogeographic framework for the dispersal of this species.

For the estimation of body mass (*West*), we used the equation of Martin et al [15], which considers the anteroposterior length of the first lower molar. This estimation was carried out in the fossil material of *Sigmodon minor* from SMA in the MPGJ (S2 Table).

Differences in molar length (m1) among *Sigmodon minor* from GTO 19, México, Borchers and Rexroad Kansas localities were evaluated using a one-way Welch analysis of variance (Welch ANOVA). Because only group summary statistics (mean, standard deviation, and sample size) were available in Peláez-Campomanes and Martin [5], pairwise differences were assessed using Welch t-tests with Holm correction for multiple comparisons rather than standard post hoc procedures. Effect sizes were estimated using Hedges' g, and uncertainty was summarized using 95% confidence intervals calculated as ±1.96 standard errors. All analyses and figures were generated using custom scripts in R.

## Institutional abbreviation

LACM, Vertebrate Paleontology Collection of Natural History Museum of Los Angeles County; LACMm, Mammalogy Collection of Natural History Museum of Los Angeles County; LEI, Laboratorio de Estudios Isotópicos, Instituto de Geociencias, Universidad Nacional Autónoma de México; MPGJ, Colección del Museo de Paleontología del Instituto de Geociencias, UNAM; UNAM, Universidad Nacional Autónoma de México.

## Results

### Systematic paleontology

Class MAMMALIA Linnaeus, 1758

Order RODENTIA Bowdich, 1821

Family CRICETIDAE Fisher, 1817

Tribe SIGMODONTINI Wagner, 1843

Genus *Sigmodon* Say and Ord, 1825

*Sigmodon minor* Gidley, 1922

Synonymy: *Sigmodon medius* Gidley, 1922

Locality and age: GTO 6 "Arrastracaballos" of early Blancan with a radiometric age younger than 3.68±0.18 Ma; GTO 12 "La Pantera" of early Blancan with a radiometric age older than 3.73±0.24 and younger than 3.9±0.3; GTO 19 "Pecos" of early Blancan.

**Referred specimens**

Maxillaries: In the locality GTO 19 "Pecos": MPGJ 2670, left maxillary with M1 and M2 (Fig 2a-c); MPGJ 2692, right maxillary with M1 and M2 (Fig 2d-f); MPGJ 2674, left maxillary with M1 and M2 (Fig 2g-1); MPGJ 2667, left maxillary with M2 and M3 (Fig 2j-k); MPGJ 2653, right maxillary with M1-M3 (Fig 2l, m); MPGJ 2672, right maxillary with M1 and M2 (Fig 2n-o); MPGJ 2675, left maxillary with M1 and M2 (Fig 2p-q); MPGJ 2689, right maxillary with M1 (Fig 2r); MPGJ 2656, right maxillary with M1 and M2; MPGJ 2661, right maxillary with M1 and M2; MPGJ 2662, right maxillary with M1 and M2; MPGJ 2663, right maxillary with M1 and M2; MPGJ 2680, right maxillary with M1 and M2; MPGJ 2683, right maxillary with M1-M3; MPGJ 2686, right maxillary with M1 and M2; MPGJ 2688, right maxillary with M1; MPGJ 2691, right maxillary with M1; MPGJ 2694, right maxillary with M1 and M2; MPGJ 2710, right maxillary with M1 and M2; MPGJ 2664, left maxillary with M2 and M3; MPGJ 2679, left maxillary with M1 and incomplete M2; MPGJ 2693, left maxillary with M1; MPGJ 2707, left maxillary with M1 and M2; MPGJ 2709, left maxillary with M1; MPGJ 2711, left maxillary with M1; MPGJ 2666, left maxillary with M1. In the locality GTO-12 "La Pantera": MPGJ 1908, left maxillary with M1 (Fig 2s).

Isolated upper molars: In the locality GTO 19 "Pecos": MPGJ 2714, right upper M1; MPGJ 2722, right upper M1; MPGJ 2724, right upper M1; MPGJ 2720, left upper M1; MPGJ 2721, left upper M1; MPGJ 2723, left upper M1; MPGJ 2727, left upper M1; MPGJ 2741, left upper M1; MPGJ 2716, left upper M1 (Fig 2t-u); MPGJ 2737, right upper M2; MPGJ 2739, right upper M2; MPGJ 2742, right upper M3; MPGJ 2713, left upper M3.

Mandibles: In the locality GTO 6 "Arrastracaballos" MPGJ 6007 (Fig 3a-c). In the locality GTO 19 "Pecos": MPGJ 2650, left lower jaw with m1-m3 (Fig 3d-f); MPGJ 2673, Left lower jaw with m1 and m2 (Fig 3g-i); MPGJ 2668, right lower jaw with m2 and m3 (Fig 3j-l); MPGJ 2669, right lower jaw with m1-m3 (Fig 3m-o); MPGJ 2651, right lower jaw with m1 and m2; MPGJ 2655, right lower jaw with m1-m3; MPGJ 2660, right lower jaw with m1; MPGJ 2678, right lower jaw m1; MPGJ 2681, right lower jaw with m1-m3 (Fig 4); MPGJ 2690, right lower jaw with m1 and m2; MPGJ 2652, left lower jaw with m2 and m3; MPGJ 2654, left lower jaw with m1 and m2; MPGJ 2671, left lower jaw with m1; MPGJ 2676, left lower jaw with I1 and M2; MPGJ 2682, left lower jaw with m1-m3; MPGJ 2695, left lower jaw with m1 and m2; MPGJ 2708, left lower jaw with m1 and m2; MPGJ 2712, left lower jaw with m1 and m2.

Isolated lower molars: In the locality GTO 12 "La Pantera" MPGJ 1907 (Fig 3p) right lower m1. In the locality GTO 19 "Pecos": MPGJ 2736, right lower m1 (Fig 3q-r); MPGJ 2717, right lower m1; MPGJ 2725, right lower m1; MPGJ 2726, right lower m1; MPGJ 2729, right lower m1; MPGJ 2730, right lower m1; MPGJ 2715, left lower m1; MPGJ 2731, left lower m1; MPGJ 2733, left lower m1; MPGJ 2732, right lower m2; MPGJ 2718, left lower m2; MPGJ 2728, left lower m2; MPGJ 2738, left lower m2; MPGJ 2735, right lower m3.

**Emended diagnosis.** *Sigmodon* species identifiable by the following set of characters: molars with rounded and well-developed cusps (ids), forming a simple occlusal surface with minimal development of accessory cusps (ids), lophs (ids), or conules (ids) (Figs 2–4); flexus (ids) are narrow and slightly oblique, with labial and lingual cuspids arranged alternately and tending to organize into lophs; occasionally, a small accessory rootlet is present in M1, M2, and m1; molars M3 and m3 are notably larger compared to other Sigmodontini; the anteroposterior length of M3 is similar to M2, and the anteroposterior length of m3 is similar to m1; Anteroloph in M2 and M3 is well-developed; Masseteric ridge terminates below the anterior root of m1, and mental foramen is positioned dorsally on the posterior diastema.

**Description**

Fragment of maxillary: The zygomatic plate is broad, and its posterior margin is at the base of the anterior M1 root (Fig 2a, 2d, 2g, 2n, 2r and 2s). Incisive foramina extend posteriorly between the anterocone and the protocone, variations that could

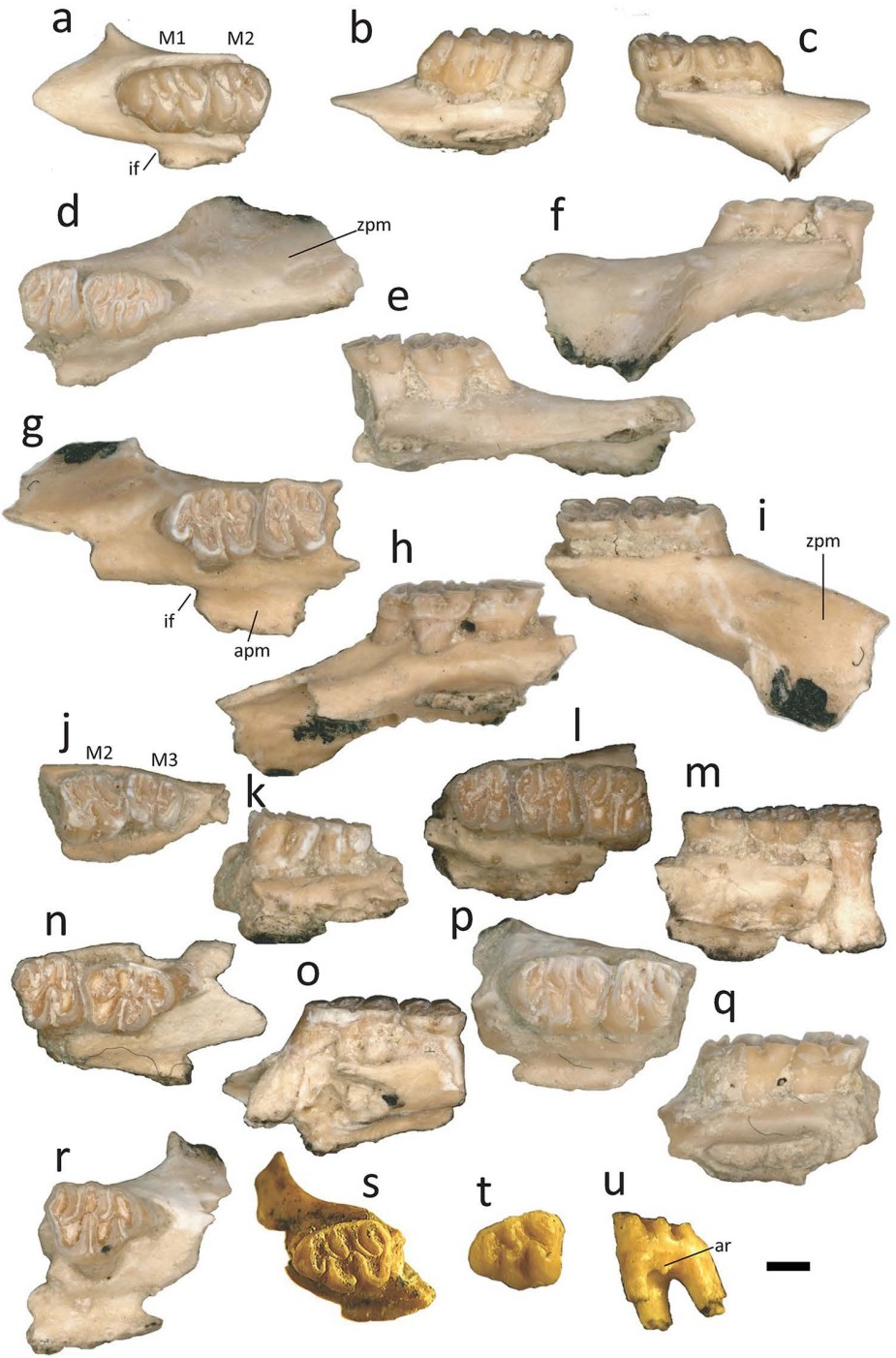

**Fig 2. Maxillaries and upper teeth of *Sigmodon minor* from the early Blancan of San Miguel de Allende Basin, Mexico. a-c)** MPGJ 2670, left maxillary with M1 and M2, occlusal **(a)**, lingual **(b)** and labial **(c)** views; **d-f)** MPGJ 2692 right maxillary with M1 and M2; occlusal **(d)**, lingual **(e)** and labial **(f)** views; **g-i)** MPGJ 2674 left maxillary with M1 and M2, occlusal **(g)**, lingual **(h)**, and labial **(i)** views; **j, k)** MPGJ 2667 left maxillary with M2 and M3, occlusal **(j)** and lingual **(k)** views; **l, m)** MPGJ 2653, right maxillary with M1-M3, occlusal **(l)** and lingual **(m)**; **n, o)** MPGJ 2672, right maxillary with M1 and M2, occlusal **(n)** and labial **(o)** views; **p, q)** MPGJ 2675, left maxillary with M1 and M2, occlusal **(p)** and lingual **(q)** views; **r)** MPGJ 2689, right maxillary with M1; **s)** MPGJ-1908, left maxillary with M1, occlusal view; **t, u)** MPGJ 2716, left upper M1, occlusal **(t)** and labial **(u)** views. Anatomical abbreviations: apm, alveolar process of maxillary; ar, accessory root; M1, first upper molar; M2, second upper molar; M3 third upper molar; if, incisive foramen; zpm, zygomatic plate of maxillary. Scale bar is 1 mm.

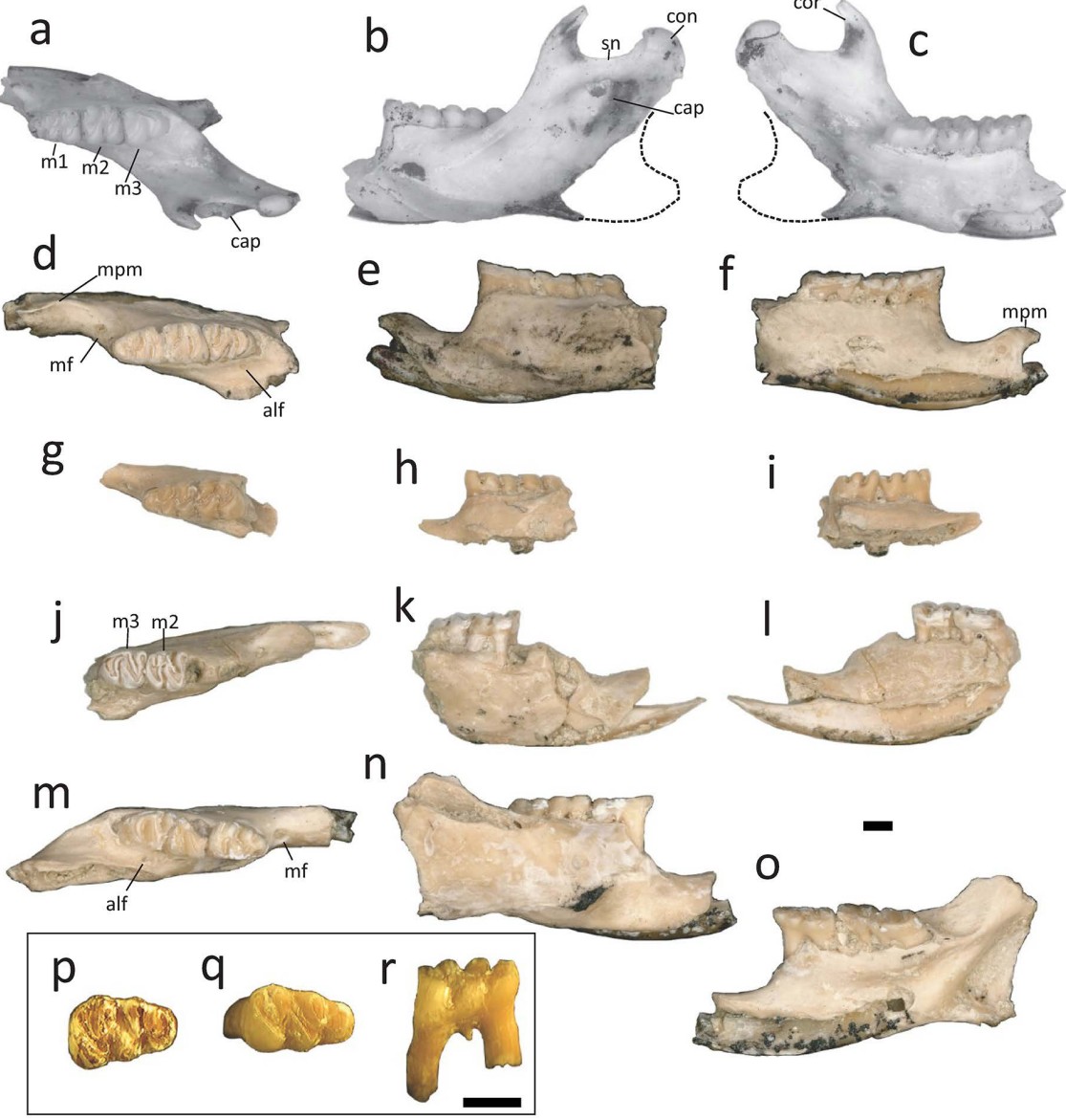

**Fig 3. Lower jaws and lower teeth of *Sigmodon minor* from the early Blancan of San Miguel de Allende Basin, Mexico. a-c)** MPGJ 6007, left lower jaw with m1 and m2, occlusal **(a)**, labial **(b)**, and lingual **(c)** views; **d-f)** MPGJ 2650, left lower jaw with m1-m3, occlusal **(d)**, labial **(e)** and lingual **(f)** views; **g-i)** MPGJ 2673 left lower jaw with m1-m3, occlusal view **(g)**, labial **(h)** and lingual **(i)** views; **j-l)** MPGJ 2668, right lower jaw with i1, m2 and m3, **j)** occlusal view, **k)** labial view, **l)** lingual view; **m-o)** MPGJ 2669, right lower jaw with m1-m3, **m)** occlusal view, **n)** labial view, **o)** lingual view; **p)** MPGJ 1907, right lower m1; **q-r)** MPGJ 2736, right lower m1, **p)** occlusal view, **q)** labial view;. Anatomical abbreviations: alf, alveolar foramen; cap, capsular process of the lower incisor alveolus; cor, coronoid process; con, condyloid process; m1, first lower molar; m2, second lower molar; m3, third lower molar; mf, mental foramen; mpm, medioventral process of mandibular ramus; sn, sigmoid notch. Scale bar is 1 mm.

depend on ontogeny based on Weksler [26], as occurs in MPGH 2670 (Fig 2a), a younger individual than MPGJ 2672 (Fig 2n). The alveolar process of maxillary is clearly enlarged (See occlusal and lingual views of maxillary elements in Fig 2).

M1: The mesodont and bunodont M1 have a mean length of 2.21 mm and a mean width of 1.68 (Table 1). The molar has five cusps: paracone, metacone, protocone, hypocone and anterocone (Fig 2a, 2d, 2g, 2n, 2p, 2r, 2s and 2t). The lingual

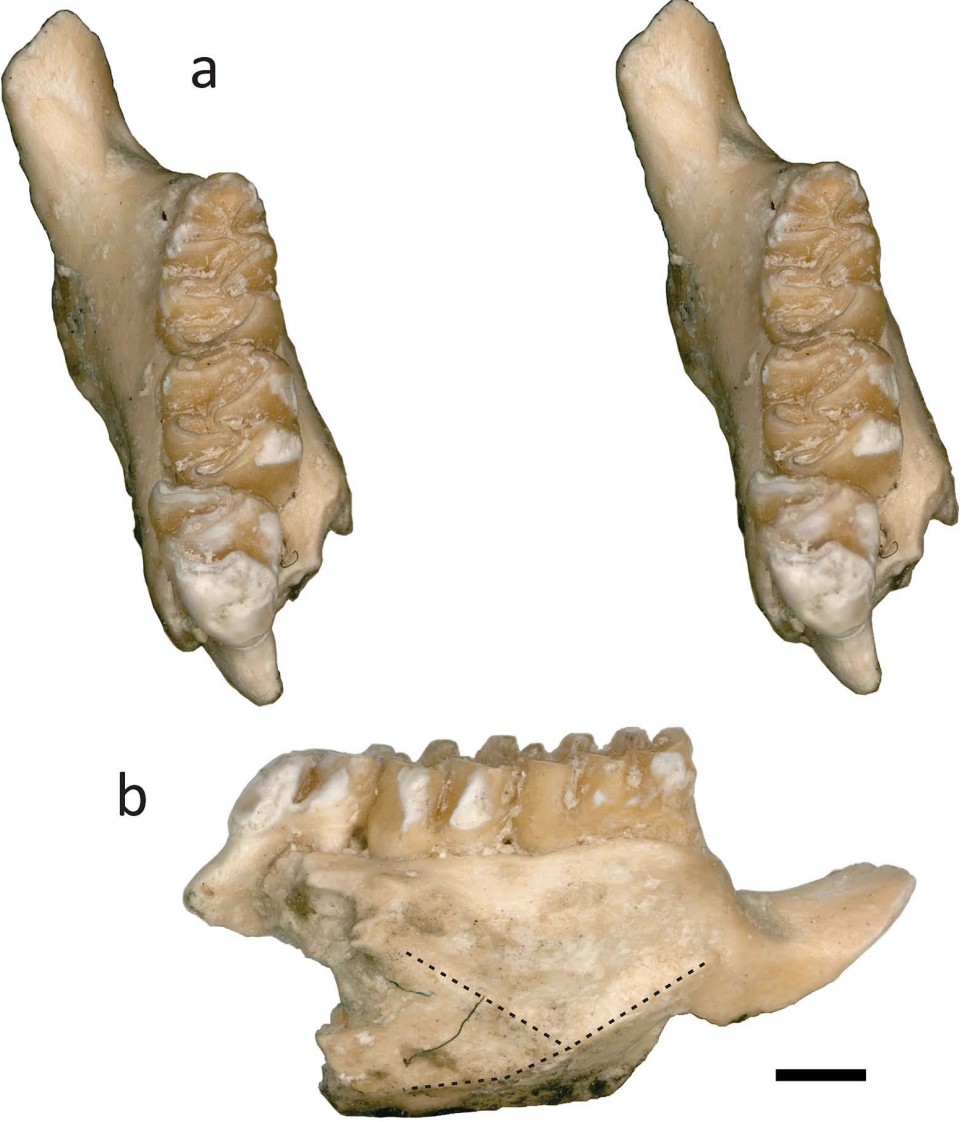

**Fig 4. Left lower jaw of *Sigmodon minor* from San Miguel de Allende Basin (MPGJ 2681). a)** stereo photos in occlusal view; **b)** labial view. The dashed line shows the ridges of the masseteric crest. Scale bar is 1 mm.

and labial cusps are nearly aligned; however, the lingual cusps are situated posterior to the labial cusps. The anterocone is elliptical and compressed, with an anteromedial groove that is almost imperceptible in young specimens (Fig 2t) and absent in mature ones The anterocone is symmetrical but oriented more labially and connected medially with the anterior arm of the protocone. All the cusps are compressed to the posterior direction in lateral views (Figs 2b, 2c, 2e, 2f, 2h, 2i, 2k, 2o and 2q). The metacone and paracone rotated more than the protocone and hypocone, and for this reason the labial cusps are more oblique than the lingual ones. The flexes are deep and narrow, and paraflexus and metaflexus tend to be oblique and parallel to each other. The hypoflexus merges with the paracone and the protoflexus joins to the base of the paraflexus, both perpendicular to the lingual margin. The metacone's posterolabial margin has a slope, but lacks a posteroloph. The molars have three main roots and an additional accessory root located beneath the paracone (Fig 2f and 2u).

**Table 1. Dental measurements (in mm) of *Sigmodon minor* from the early Blancan of San Miguel de Allende Basin, central Mexico. Abbreviations: M1-M3) upper molars; m1-m3) lower molars; N, number of specimens.**

| Locality | Element | N | Length mean | Length range | Width mean | Width range |
|---|---|---|---|---|---|---|
| GTO 19 | M1 | 32 | 2.21 | 1.73–2.5 | 1.68 | 1.42–1.89 |
| | M2 | 21 | 1.53 | 1.38–1.85 | 1.64 | 1.35–1.86 |
| | M3 | 5 | 1.37 | 120–1.57 | 1.34 | 1.14–1.73 |
| | m1 | 24 | 2.15 | 1.95–2.39 | 1.44 | 1.31–1.58 |
| | m2 | 16 | 1.67 | 1.5–1.84 | 1.62 | 1.51–1.76 |
| | m3 | 10 | 1.88 | 1.52–2.06 | 1.54 | 1.39–1.65 |
| GTO 6 | m1 | 1 | 2.24 | | 1.61 | |
| | m2 | 1 | 1.82 | | 1.66 | |
| | m3 | 1 | 1.95 | | 1.62 | |
| GTO 12 | M1 | 1 | 1.94 | | 1.62 | |
| | m1 | 1 | 2.22 | | 1.40 | |

M2: The mesodont and quadrate M2 have four cusps: paracone, metacone, protocone and hypocone (Fig 2a, 2d, 2g, 2j, 2l, 2m and 2p). The cusps are similar in form and position to M1, without an anterocone. Instead, there is an anteroloph that is connected to the protocone. The absence of the anterocone allows for a greater development of the protocone, which is similar in size to the hypocone. In some specimens a small paralophule is present (Fig 2a and 2n). The molars have three roots.

M3: The mesodont and bunodont M3 have four cusps: paracone, metacone, protocone, and hypocone (Fig 2j and 2l). The anteroloph is well developed and is fused with the protocone, forming the anterior loph. In the posterior region of the molar, the paracone connects with the hypocone and this connects with the metacone. Enlargement of the anteroloph altered the shape of the molar, giving it a trapezoidal appearance. The major cusps are less compressed than those on M1 and M2, and the flexes are narrow and perpendicular to the anteroposterior edge of the molar. The metaflexus is L shaped, with the shorter side directed to the posterior region. The molars have three roots.

Mandibular remains: The jaws have an m3 of a similar size to m1 (Figs 3a, 3d and 3m, 4). The mental foramen is in the dorsal position of the diastema at the base of the anterior root of the m1 (Figs 3d, 3g, 3j and 3m, and 4). The masseteric crest is composed of two ridges that are fused together beneath the hypocone of m1 or the boundary between m1 and m2, and extends in the direction of the mental foramen (Fig 3b, 3e and 3n). The medioventral process of the mandibular ramus is weakly present (Fig 3d and 3f). The ascending ramus is elongated and stars in the limits of m2 and m3, almost covering the totality of the crown of m3 in a labial view (Fig 3b and 3n). The coronoid process was nearly at the same level as the condyloid process, curving posterodorsally with a nearly vertical orientation (Fig 3b and 3c). The sigmoid notch is open and rounded (Fig 3b and 3c). The capsular process of the lower incisor alveolus is well developed with an acute projection posterior to the sigmoid notch (Fig 3b and 3c); this character is highly variable and could depend on the ontogeny according to Weksler [26]. The lower incisor has a straight lake of dentine on the occlusal surface (Fig 2j).

m1: The mesodont and pentalophodont m1 have five cuspids: metaconid, entoconid, protoconid, hypoconid and anteroconid (Fig 3a, 3d, 3g, 3m, 3p and 3q). The lingual and labial cuspids are not aligned, and the labial cuspids are situated anterior to the labial cuspids. The anteroconid is simple, semi-circular, and not compressed anteroposteriorly; it is symmetrical and evenly placed between the labial and lingual sides, and is connected to the metaconid in the middle. In young specimens the anterior murid is not well-formed, in this case the anteroconid is almost disconnected and the protoflexid and metaflexid may be connected (Fig 3g and 3p). All cuspids are globular and inclined towards the anterior region, where the highest point of the crown is observed. The posterolophid is highly developed and the ectostylid is present. The flexids are deep and narrow, the metaflexid and posteroflexid tend to be oblique and parallel to each other but shorter

than the entoflexid. The hypoflexid is directed towards the entoconid perpendicular to the labial margin. The molars haves two main roots and occasionally a small accessory root between them (Fig 3r).

m2: The mesodont and quadrate m2 have four cusps: metaconid, entoconid, protoconid, and hypoconid (Fig 3a, 3d, 3g, 3j and 3m.) These cusps have a similar form and position to those of the M1, but without an anteroconid. The posterolophid is well developed. The absence of the anteroconid does not have much influence on the anterior region of the molar, as in M2, but a well-developed anterolabial cingulid and protoflexid are observable (Fig 3j). The molars have two roots.

m3: The mesodont and bunodont m3 have four fused cuspids forming an S-shaped outline: metaconid, entoconid, protoconid, and hypoconid (Fig 3a, 3d, 3j and 3m). On the anterior side, the metaconid is connected with the protoconid, and the protoconid is connected with the entoconid. The entoconid is reduced in size due to the compression of the hypoflexid and is connected with the posterior lophid, which is formed by the hypoconid and posterolophid. Only two deep flexids are dividing the cuspids, entoflexid and hypoflexid. The anterolabial cingulid is present, as it is in the m2. The molars have two roots.

## Discussion

### Anatomical discussion

These specimens of *Sigmodon minor* represent one of the most comprehensive populations of this species reported to date. The assemblage includes well-preserved maxillary and mandibular elements, with molars exhibiting two or three roots, providing important morphological information for taxonomic and comparative analyses. The observed variation in molar wear is interpreted as evidence of ontogenetic changes associated with aging, offering valuable insights into dental anatomical changes throughout the lifespan of these rodents. Furthermore, this study documents previously unknown or poorly understood cranial and mandibular traits in detail, substantially enhancing our understanding of their morphological variability. Altogether, these findings significantly advance knowledge of the evolutionary history and paleoecology of this extinct species.

The molars of *Sigmodon minor* are notable for their morphological stability, despite being a species that was widely distributed throughout Pliocene of North America, from California to Florida and from Nebraska to central Mexico [5,14,27,28]. This species has a simple occlusal structure, which made its taxonomic identification easier, but in larger populations and longer time periods some variations and changes in size have been documented [5,15]. Based on these previous studies and using the fossil material described here for SMA, it is possible to discuss morphological variations on: simple occlusal structure, shape of anterocone (id) and accessory roots in m1.

It is important to note that these comparisons should be made in juvenile organisms or with minimal wear on the cusp level. As the wear on the molar increases, it becomes difficult to distinguish between species of *Sigmodon*, including *Prosigmodon*. Additionally, the major cusps in the molars have a conical development towards the base, and their shape, obliquity, type of connection, and width of the flexus (ids) depend on their ontogenetic or wear state. This can be seen by comparing the juvenile specimen MPGJ 2670 (Fig 2a-c) with the more worn specimens MPGJ 2692 and MPGJ 2674 (Fig 2d-i). As can be seen, the obliquity of the cusps changes with wear, and there is an increase in the proportion that the anterocone occupies in the molar. To this end, we have taken late ontogenetic condition into account in the following discussions.

### Simple pattern in molars

*Sigmodon* species are characterized by a relatively simple occlusal structure, marked by prominent cusps and a rare presence of accessory features. This simplicity distinguishes *Sigmodon* from species of *Prosigmodon* [10] and South American sigmodontines [26], which present more complex occlusal surfaces. Only *Rheomys*, the sister genus of *Sigmodon*,

appears to exhibit similar characteristics, sharing a tendency towards simplified pattern on its occlusal surfaces. This simplification may be linked to the development of hypsodonty, where molars with higher crowns are also adapted into a simpler, more lophate structure, a condition potentially driven by heterochronic processes [29].

The occlusal simplicity in the molars of *S. minor* is evident in its earliest records from the Meade Basin, and this trait persists in the material found in the SMA basin. Although accessory cusps (ids) are absent on the molars, but there is a gain in space by an anteroloph and a posteroloph depending on the position of the molars, where the upper molars M2-M3 develop an anteroloph, while the lower molars m1-m3 develop a posteroloph (Figs 3 and 4). This characteristic, also observed in modern species, distinguishes *Sigmodon* from *Prosigmodon*, where these structures are noticeably reduced in size and complexity.

In lower molars, the lingual cingulid is a well-preserved feature, which deepens towards the anterior region. For this reason, the anterior cingulid in m2 and m3 exhibits considerable depth, consistent with characteristics seen in modern species (Fig 4). Notably, in certain SMA specimens a small ectostylid was observed developing on the labial cingulid (MPGJ 2673, Fig 3g), a distinctive feature not shared with any other *Sigmodon minor* variant or *Sigmodon* species.

The simplification of structures is also evident in M3 (MPGJ 2667, Fig 2j), displaying a significant reduction in the posteroloph, making it nearly indistinguishable from the hypocone. However, an invagination of the enamel, representing the remnant of the posteroflexus, can still be discerned in its connection with the metacone. This simplified condition is consistent with modern *Sigmodon* species, even in young specimens, a noticeable reduction in the posteroloph can be observed, as exemplified by *Sigmodon hispidus* (LACMm 4688, in S2 Fig). It's worth noting that this feature is also observed in *Prosigmodon tecolotum* (MPGJ 3336, Fig 4c in Pacheco-Castro [11]) and *P. ferrusquiae* [IGCU 7358 and 7361, Plate 3 in Carranza-Castañeda and Walton [9]], setting them apart from *Prosigmodon holocuspis* [Fig 5 in Peláez-Campomanes and Martin [5]], where this structure remains well developed.

In addition to the loss of accessory cusps in the evolution of *Sigmodon* Peláez-Campomanes and Martin [5] describe a trend where molars modify the bulbous shape of cusps into hypsodont lophs, anteroposteriorly constricted. In this regard, the fossils from SMA exhibit pleisiomorphic characters, such as cusps being conical and bulbous at the base, and the anterior and posterior walls tend to be inclined (see lateral views in Figs 2–4). This anteroposterior compression of cusps is progressive in more recent species (see S2 Fig). Therefore, the observed hypsodonty in *S. minor* represents an early stage in the evolutionary process leading to the development of higher crowns.

It is a conventional notion to regard the second molars (M2 and m2) as a simple repetition of the first molars (M1 and m1), lacking the anterocone (id). However, upon examining complete upper or lower jaws, distinctions between these molars can be observed. For instance, in specimens MPGJ 2670 (Fig 2a) and MPGH 2672 (Fig 2n), the paracone of M2 develops a paralophule, a feature absent in M1. Interestingly, this structural variation does not seem to correlate with the degree of molar wear but rather appears as a characteristic in specific individuals. According to Martin et al [23], the presence of paralophules in M1 is diagnostic for identifying extinct members of the Sigmodontinae subfamily, particularly oryzomyinins, while this structure remains absent in phyllotinins and sigmodontins. In the review of contemporary *Sigmodon* species conducted here, this distinctive structure has been rarely observed, but in some young individuals the paralophule is barely distinguishable (e.g., LACMm 12274, M2 of *Sigmodon fulviventer*; LACMm 86676, M2 of *Sigmodon hispidus texianus,* in S2 Fig). This may suggest that the loss of these structures initially impacts the first molars before affecting the second molars, a hypsodonty process that might act differentially based on the molar position.

In identifying *Sigmodon* species, another characteristic under consideration is the width of the reentrants, which may distinguish *Sigmodon curtisi* from *S. hudspethensis* [30]. Interestingly, there are similarities in the shape and size of *Sigmodon minor* material from Mexico with *S. hudspethensis*, especially regarding the shape and width of the reentrants, a feature that has been critical in differentiating this species. However, this differentiation is likely not feasible during juvenile stages [31]. Within the material examined here, variation in reentrant width is observable, but a clear boundary to separate this variation could not be established. Notably, when comparing specimens such as MPGJ 2736 (Fig 3q) and MPGJ

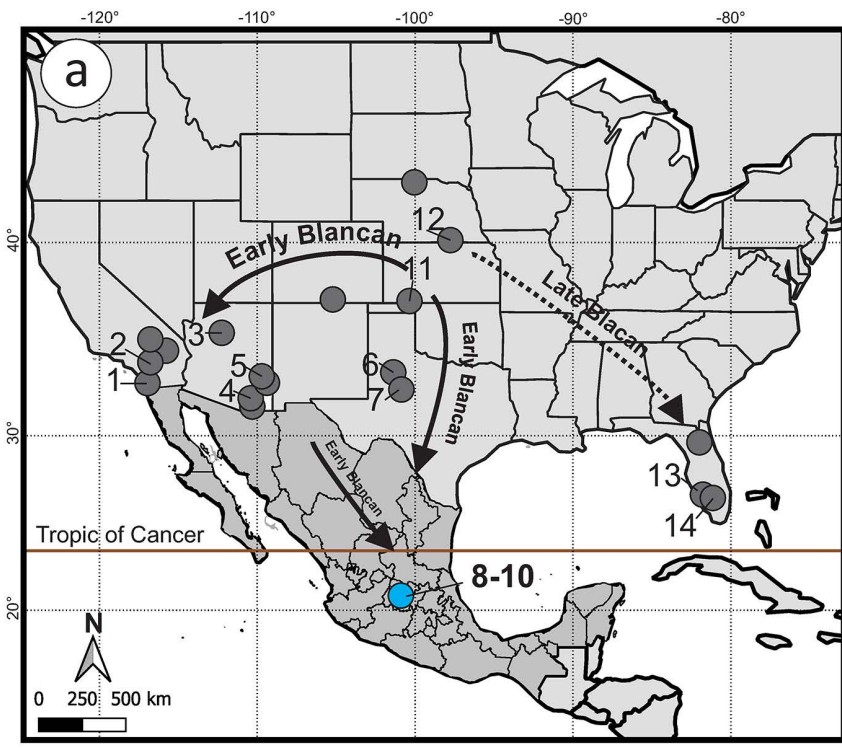

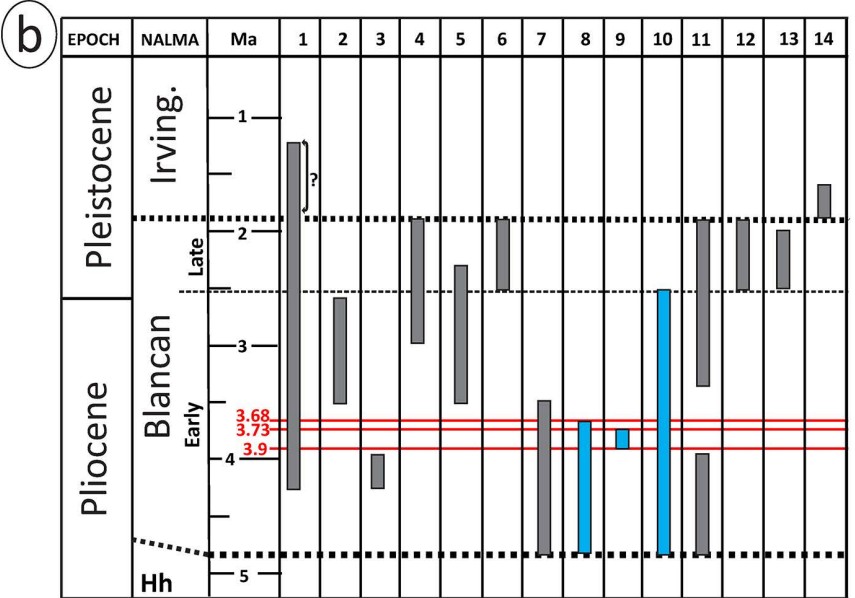

**Fig 5. Map of distribution and stratigraphic record of *Sigmodon minor* in North America.** A, map of distribution of *S. minor* that includes the previous records obtained from the Paleobiology Database (Paleodbd.org, revision January 6 of 2025, see S2 Table) in grey and the new records from SMA basin in blue. B, chronologic ranges of *S. minor* in the principal localities of North America. The columns are ordered in direction east-west, the grey color are previous records and in blue the new records in this study. Code of the numbers refer the following formations, basins and localities: 1, Palm Spring Formation; 2 San Timoteo Formation; 3, Verde Formation; 4, St. David Formation; 5, Gila Conglomerate; 6, Blanco Formation; 7, Beck Ranch Fauna; 8, GTO 6 "Arrastracaballos"; 9, GTO 12 "La Pantera"; 10, GTO 19 "Pecos"; 11, Meade Basin; 12, White Rock Fauna; 13, Pinecrest Beds. Abbreviations: Hh) Hemphillian; Irving) Irvingtonian; Ma) millions of years before the present; NALMA) North American Land Mammal Ages;?) doubts in the record.

2681 (Fig 4), differences in the entoflexid opening are evident. These morphological variations likely arise from ontogeny and their correlation with the anteroposterior length of the molar, as elongated molars tend to exhibit these features.

Furthermore, the molar size in the SMA material exceeds that of *Sigmodon minor/medius* and closely resembles *S. hudspethensis*. The lower molars identified as *S. minor* lack key characteristics typical of *S. hudspethensis*, such as a well-defined labial root and greater complexity in the peg-like labial roots [30]. Additionally, *S. hudspethensis* populations are dated to the late Blancan, raising intriguing possibilities for future research to explore potential connections between central Mexican *Sigmodon minor* populations and these later rodent variants from Texas.

### Anterocone (id)

In the sigmodontines, the anterocone (id), also known as the procingulum [22], is a highly variable structure with significant diagnostic importance. This structure forms a complex of minor cusps, which are highly diagnostic for identifying rodents within the subfamily Sigmodontinae [32]. However, in *Sigmodon* species, there is a tendency for these cusps and conules to merge into a single cusp with variable symmetries. It is possible that the development of this cusp (id) is influenced by a morphogenetic process involving the establishment of well-developed enamel knots [33]. However, this phenomenon is not well understood, and a more detailed study on the development of this cusp could help determine whether it represents a case analogous to the evolution of tribosphenic molars into herbivorous forms, as exemplified by the emergence of the hypocone. For this reason, and due to the significance of this cusp (id), we employ the term anterocone (id) to refer to species of *Sigmodon*. Notably, this genus does not exhibit the evolutionary trends seen in other Sigmodontinae species (particularly South American species), where the complexity of the multicuspid structure in the anterior region of the first molar is more dynamic and can be identified as a procingulum, according to Barbière et al. [24].

The anterocone (id) is a diagnostic structure in the North American Sigmodontini lineage, and its shape helps us distinguish between species of *Sigmodon* and other genera, such as *Copemys* or *Prosigmodon*, in which this structure is notoriously bilobed [10]. However, in the oldest *Sigmodon minor* material from the Meade Basin, this structure was described as bilobated [5]. The anterocone of the *S. minor* of Mexico is simple-cusped with just a couple of exceptions, like in MPGJ 2716 (Fig 2t) where it is possible to see a small anteromedial groove. However, it is not possible to distinguish clearly the conules (ids) in the majority of specimens. In some very young specimens of recent species, the anteroconid can barely recognized (*S. arizonae*, LACMm 11466, S2 Fig). In contrast, all the materials described in this work presented a simple anteroconid, which may even have a trefoil shape, expanded towards the anterior region, similar to the anteroconid that can be observed in some specimens of *S. mascotensis* (LACMm 12273, S2 Fig). Based on our observations, we can infer that SMA cotton rats represent a more derived morphotype than the one described in the early Blancan section of the Great Plains.

Another aspect to consider regarding the anterocone (id) of *Sigmodon minor* is its trend to be symmetrical and to be medially connected with the protocone (id) (Figs 2–4), a character shared with older species and varieties of *Sigmodon* [5]. This is a distinctive feature compared to modern species, where a medial connection of the anterocone with the protocone is observed in the lower molars, but in the upper molars it can vary, the connection being sometimes medial and sometimes labial (See S2 Fig and compare lower and upper teeth).

### Accessory roots

The number of roots in the m1 is a key characteristic for distinguishing older populations or varieties of rodents from younger ones of *Sigmodon minor* in the Great Plains. In the stratigraphic sequence from the early to late Blancan, the earliest varieties, described by Peláez-Campomanes and Martin [5] as "*Sigmodon minor/medius*" (previously classified as *Sigmodon medius*), may possess a single accessory root. In contrast, the smaller and more derived variety, "*Sigmodon minor/minor*", exhibits an increased number of roots, often referred to as "tiny pegs." Moreover, the most derived species, such as *S. curtisi*, typically feature three roots on each lower molar, while *S. hispidus* presents four [5,30].

In the *Sigmodon minor* specimens from the SMA, only a small accessory root was observed on both upper and lower molars (MPGJ 2716, Fig 2u; Fig 3r). This observation relates the populations of GTO 6, GTO 12, and GTO 19 with the *Sigmodon minor/medius* varieties from the Meade Basin, although with a notable difference: the SMA specimens exhibit a larger size. Generally, early Blancan varieties of *S. minor* have been described as possessing at least one accessory root on the m1. However, archaic forms from the Meade Basin [5] and records from the Verde Formation [34] lack any root stems. Based on the evolutionary stages proposed by Peláez-Campomanes and Martin [5], the SMA population represents a derived stage of these rodents, suggesting closer affinities with populations from the Great Plains rather than those from western North America. A direct comparison with material from the Beck Ranch Local Fauna [14] would be particularly insightful, as this record is contemporaneous with those from central Mexico. Unfortunately, it remains undescribed, with only a brief mention that the assemblage consists of "many jaws and hundreds of isolated teeth".

## Asymmetrical geographic dispersion of *Sigmodon minor* in North America

The new record of *Sigmodon minor* in central Mexico represents a significant extension of its known distribution, reaching over 1500 km southward into areas beyond the Tropic of Cancer. This finding enhances our understanding of the faunal succession during the transition from the late Hemphillian to the early Blancan in central Mexico. The association of *Nannippus peninsulatus* and *Plesippus simplicidens*, along with radiometric dating, provides precise chronological constraints that strengthen regional and continental biochronological correlations [9,11]. Despite the relevance of this record, early Blancan rodent faunas in Mexico remain scarce, largely due to the limited number of well-sampled localities.

Interestingly, *Sigmodon minor* is absent in other Blancan Mexican localities where screen-washing techniques have been applied. For example, Lindsay and Jacobs [8] found no evidence of this genus in La Concha, Chihuahua, despite intensive sampling and the presence of related genera such as *Copemys* and *Prosigmodon*, as well as other rodents of northern origin such as G*eomys minor* or *Pliophenacomys willsoni*. This suggests potential ecological or biogeographical barriers that may have restricted the southward expansion of *Sigmodon* into northern Mexico during the Blancan, especially considering that faunal similarities between Yepómera and central Mexican localities had previously been reported [9].

In contrast, numerous Blancan localities in the United States offer a detailed temporal and spatial record of *Sigmodon minor* (Fig 5). The Meade Basin in Kansas presents the oldest known occurrences, with *S. minor* occurring in the Fallen Angel B locality between 5-4.5 Ma [5]. However, in the work of Martin et al. [15], this locality is related to geomagnetic polarity chron Gilbert C3n.2r with an age between 4.80 and 4.62 Ma, which is the age considered in this work. The species is nearly continuous throughout the Blancan in this region (except during the 4.18 and 3.33 Ma in XIT and Hornet localities), with its last record in the Borchers locality dated to 2.10 Ma, based on the Huckleberry Ridge ash [35]. These data support an origin of this species in the Great Plains followed by rapid and asymmetrical dispersal trough North America.

In the western United States, *Sigmodon minor* is documented in the Palm Spring Formation of California, along more than 2 km of the section in the Diablo-Olla and Tapiado-Huesos Members, with its earliest presence dated to ~4.29 Ma, based on the Layer Cake fauna, and potentially persisting until ~1.2 Ma in Collecting Unit 60 (Fig 4b in Cassiliano [27]). This would be the most recent time range known for *Sigmodon minor*, although a detailed review of the Irvingtonian material from Vallecito Creek is necessary to recognize the taxonomic identity of this material. A similar early occurrence is observed in the San Timoteo Formation, where the lowest stratigraphic datum corresponds to the top of Unit 3 with an age of 3.5 Ma, while the youngest occurrence is reported at the base of Unit 4 of the Gauss-Matuyama reversal, at an age close to 2.58 Ma [6]. Next to California, in Arizona, the Verde Formation also provides well-constrained early Blancan records, with localities dated to 4.26 and 3.98 Ma based on Nunivak and Cochiti subchrons respectively [34,36]. Although this stratigraphic relationship changed in the St. David Formation, San Pedro Valley, where Johnson et al. [37] reports *Sigmodon minor/minor* and *Sigmodon minor/medius* in the Benson Fauna and Curtis Ranch Fauna, with an age inferred

with magnetostratigraphic of 3.1 Ma in its maximum (Post Ranch locality), and a minimum range less than 1.71 Ma (Glyptotherium locality during the Olduvai event). According to Lindsay et al. [38], the fauna of the Curtis Ranch, where the Glyptotherium locality is, could correspond to the late Blancan and not to the early Irvingtonian, which is the basis in our adjustment of its temporal range (Fig 5). Further in the Gila Conglomerate preserves *S. minor* from 3.5 to 2.3 Ma, showing continuity across much of the Blancan [39].

Towards the center of the continent, in Texas, *Sigmodon minor* was reported by Dalquest [14] to be abundant in the early Blancan of Beck Ranch Local Fauna. This age was inferred from a complete comparison of fauna, in which fossils such as *Borophagus diversidens* and *Nannippus beckensis* stand out. According to the previous author, this fauna should be older than 3.5 Ma, based on the comparison of faunas with the Hagerman Local Fauna. Unfortunately, there are no more recent works to corroborate this assertion with magnetostratigraphy or radiometric dating, but here we tentatively accept it. The Blanco Local Fauna, type locality of the Blancan NALMA, dated to 1.77–1.4 Ma, offers a well-defined late Blancan record [40,41]. Additional support comes from the White Rock Fauna in Kansas, where molar specimens display transitional morphologies between *S. minor* and *Sigmodon minor/medius*, indicating the survival of archaic traits into the late Blancan [42].

To the east, *Sigmodon minor/medius* reaches the Pinecrest Beds of Florida (2.5–2.0 Ma) and the early Irvingtonian De Soto Shell Pit (~1.6 Ma) [7,28,43]. These marginal occurrences mirror the prolonged survival of the species in the Atlantic and Pacific region and highlight a consistent pattern: *S. minor* persists longer along the continental margins, while its central range is more temporally constrained. This observation challenges the use of *S. minor* as a Blancan index fossil outside the central Great Plains, as previously proposed by Bell et al. [41].

Altogether, the paleobiogeographic pattern of *Sigmodon minor* reflects an asymmetrical dispersion from a central origin, with early establishment in the Great Plains and subsequent dispersal into the western U.S. and central Mexico and latest to eastern U.S.

## Paleoecology and body size of *Sigmodon minor*

The wider geographical range of *Sigmodon minor* during the Blancan is attributed to the expansion of open ecosystems in North America, which resulted from the decline in temperature and increase in aridity during the Neogene period. During this time, forests reduce their distributions, and large areas of the northern hemisphere begin to be replaced by savannahs and grasslands [2,3,44].The record of this species is interpreted as a palaeoecological indicator of less extreme conditions in open ecosystems, preferentially distributed in subtropical to warm-temperate environments [6]. This pattern is evident in localities such as Beck Ranch, Arizona, where Dalquest [14] proposed that *S. minor* was a species adapted to humid conditions. He described this area as resembling an oasis or marsh within the prairies, where rodents like cotton rats, *Baiomys*, and *Ogmodontomys* thrived alongside *Nannippus beckensis*, frogs, lizards, snakes, and pond turtles, while large herbivores were notably rare.

In the SMA basin, *Sigmodon minor* is associated with vertebrate microfauna that supports the presence of wetter conditions mixed with grassland microfauna and megafauna. As occurs in GTO 12 La Pantera, where *S. minor* occurs with the rodents *Paraneotoma sawrockensis* and *Bensonomys* sp., and material of amphibians such as frogs and salamanders, as well as the equids *Nannippus peninsulatus* and *Plesippus simplicidens* (Fig 1). Similarly in GTO 6 "Arrastracaballos" this species is found in association with *Baiomys* sp. and abundant material from pond and land turtles, but mixed with grassland members such as *Paenemarmota barbouri*, *Spermophilus* sp., *Neotoma* sp. and lagomorphs, as well as the megafauna *Borophagus diversidens*, *Miracinonyx studeri*, *Nannippus peninsulatus*, *Plesippus simplicidens* and peccaries (Fig 1).

This mixture of faunas with different ecological affinities was perhaps favored by a topographically complex landscape in the sedimentary basins and river systems of central Mexico, where climatic conditions were not so extreme. In general, Sigmodontini are very sensitive to a drop in temperature, preventing them from reproducing or generating fat stores to

survive the winter [45], and their distribution is concentrated far from boreal latitudes. It has only recently been observed that the historical distribution range of *Sigmodon* has increased due to the effects of global warming, with the northern-most records of this species now extending to southern Nebraska. This is a significant change from the 1950s, when such an expansion was not possible [46].

*Sigmodon minor* experienced some morphological changes in response to substantial climatic fluctuations, especially during the colder phase of the Pliocene-Pleistocene epochs. These changes resulted in a decrease in the species size and a corresponding increase in its abundance. For example, Martin [30] proposed that the presence of accessory roots in the lower molars of variants of *Sigmodon minor/medius* and *Sigmodon minor/minor* may correspond to the transition in the feeding habits of this lineage of rodents, changing from granivorous-browsing to a pastoral grazing. Consequently, the specimens unearthed at the localities of SMA basin are more indicative of a browser diet than a pastoral grazing one.

The climatic sensitivity of *Sigmodon minor* and its entire lineage has been examined through the analysis of its varieties and species relationships. For example, Albright [6] documented in the San Timoteo Formation that this species disap-peared in the late Blancan, after the basin evolved towards cooler, cloudy, and wet conditions, such as the Mediterranean climate. While Martin and Prince [47] in the Palm Spring Formation observed that *S. minor* is replaced by a large rodent called *Sigmodon lindsayi*, which could be related to a modification in the habitat under fluctuating regional conditions. In the same way, in the Meade Basin the *Sigmodon minor/medius* morphotype is replaced by a smaller variety such as *Sig-modon minor/minor*, between the early to late Blancan [5].

Based on previous studies, differences between older and more recent populations of *Sigmodon minor* can be assessed through body size variation, as this species tends to decrease in size over time or to be replaced by larger cotton rat species in the Great Plains [5,15]. In this context, a Welch one-way ANOVA revealed significant differences in molar length among *S. minor* populations from the Rexroad (Kansas), GTO 19 (Mexico), and Borchers (Kansas) localities (p<0.001). Pairwise Welch t-tests with Holm correction showed that the Early Blancan populations from Rexroad and GTO 19 exhibit significantly larger molar lengths than the Late Blancan population from Borchers, with moderate to large effect sizes (Hedges' g = 0.58–1.55) (S3 Fig).

In the SMA Basin, *S. minor* m1 length ranges from 1.95 to 2.39 mm (Table 1), closely overlapping with values reported for *Sigmodon minor/medius* from the Meade Basin, where m1 length ranges between 1.75 and 2.36 mm. Although the Mexican sample shows a slightly larger mean m1 length (2.15 mm for GTO 19), it remains comparable to contemporane-ous Great Plains populations from Sanders (2.09 mm), Deer Park (1.98 mm), and Rexroad (2.03 mm).

We interpret these size affinities as additional evidence linking the SMA cotton rat population to older Great Plains pop-ulations, likely reflecting the persistence of *S. minor* under less cool climatic conditions in central Mexico during the Early Blancan (Fig 6).

It is noteworthy that the record of *Sigmodon minor* in the SMA basin coincides with its disappearance in the Meade Basin, between the 4 and 3.5 Ma, and it is briefly replaced by *Prosigmodon holocuspis* (Fig 6). The reasons for this are not well understood, but it is possible that this is related to a not well-defined climatic process [5,15]. Ongoing microver-tebrate prospecting within early Blancan deposits, combined with a detailed stratigraphic analysis of the SMA basin and other Mexican basins, is essential for improving our understanding of the evolutionary history, dispersal dynamics, and paleoecology of this rodent lineage, as well as its broader paleobiogeographic implications.

## Conclusion

The record of *Sigmodon minor* in the Blancan basin of SMA extends the distribution of this species to central Mexico, being the southernmost record known in North America. This record has faunal associations and radiometric dating by vol-canic ash that restrict the occurrence of these organisms to the early Blancan, this being the fourth oldest record known in North America, contemporary with faunas from the lowest levels of Palm Spring Formation in California, Verde Formation in Arizona, Beck Ranch Fauna in Texas, and the Meade Basin in Kansas. The increase in the geographic distribution of

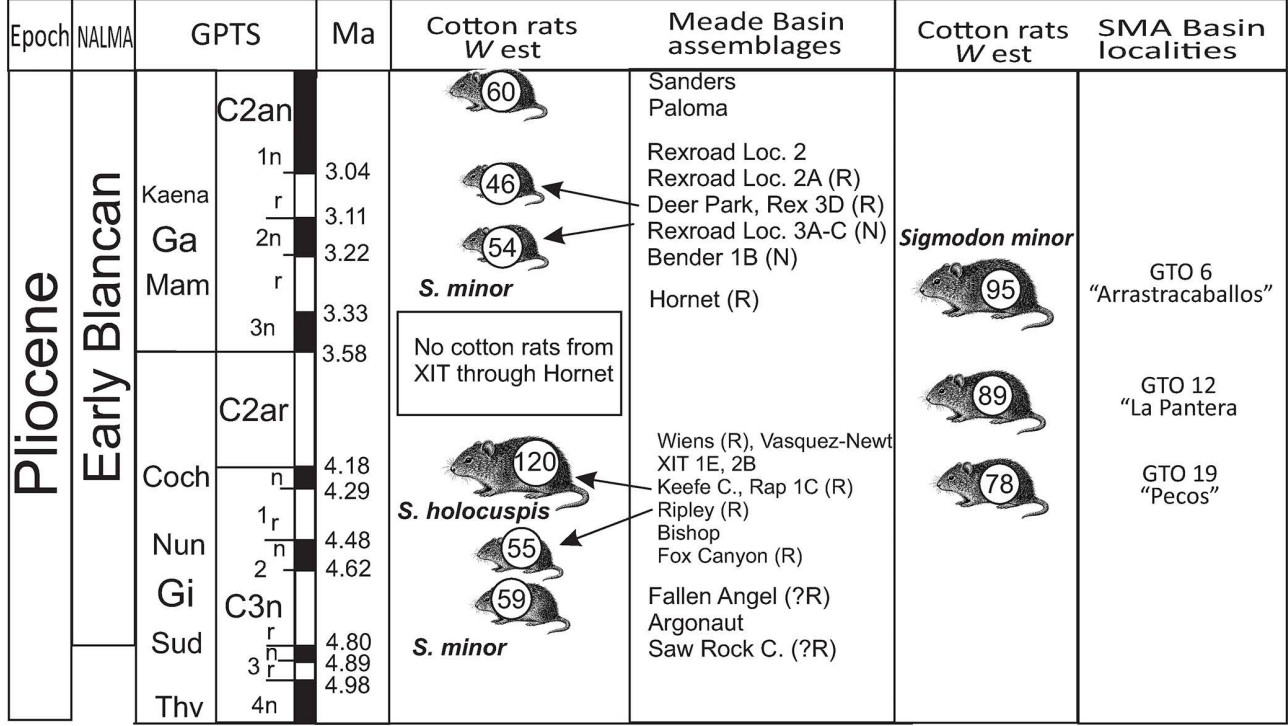

**Fig 6. Body mass estimation in cotton rats from Meade Basin, Kansas and San Miguel de Allende Basin, central Mexico.** The values and graph of Meaden Basin was taken from Martin [15] and was modified to include the records in central Mexico. Abbreviations: GPTS) Geomagnetic Polarity Time Scale; NALMA) North American Land Mammal Ages; Ma) millions of years; *W* est) body mass estimated.

*S. minor* indicates the ability of these rodents to establish themselves in much of North America and their position as an important taxon for recognizing early Blancan in Mexico.

Our new materials are one of the most complete known, which has made possible to describe anatomical structures related to mandibular and maxillary elements that are previously little known from other locations in the USA, which allows for a better understanding of population variation, ontogeny, and wear on the surface of dental elements. Furthermore, morphological comparisons of the occlusal surface of the molariform elements, especially the M1s and m1s, show that the SMA material has a simple occlusal structure with a tendency to maintain mesodont molars with slight rotation of the labial (upper) and lingual (lower) cusps. Furthermore, the majority of m1s have a small accessory root, which is considered a primitive stage in the evolution of these rodents.

We propose that the population of cotton rats in the SMA Basin may be linked to the dispersal of these rodents from the Great Plains of North America during the early Blancan. This phenomenon could potentially have been influenced by the global decrease in temperature and the expansion of grasslands. It involved a diverse assemblage of small mammals, with browsing rodents seeking favorable conditions to settle in central Mexico.

## Supporting information

**S1 Appendix. Radiometric ages of the localities in the San Miguel de Allende Basin, Guanajuato, Mexico.**
(PDF)

**S1 Fig. Dental terminology used in this study.**
(PDF)

**S2 Fig. Upper and lower molars of modern species of *Sigmodon*.**
(PDF)

**S3 Fig. Mean m1 length of *Sigmodon minor* populations from the Rexroad (Kansas), GTO 19 (central Mexico), and Borchers (Kansas) localities.**
(PDF)

**S1 Table. Records of *Sigmodon minor* of Paleobiology Database January 6, 2025.**
(PDF)

**S2 Table. Estimation of body mass (*West*) of *Sigmodon minor* from the early Blancan of San Miguel de Allende basin, Guanajuato, Central Mexico.**
(PDF)

## Acknowledgments

We are grateful to the Universidad Nacional Autónoma de México for their academic and administrative support, particularly the Laboratorio de Paleontología of Instituto de Geociencias, where the described fossils are housed. Special thanks go to Biol. Ilda Troncoso for her assistance in curating the material and to the late Mr. Harley Garbani, who collected this material alongside Oscar Carranza. We extend our gratitude to Samuel McLeod for his support in the Vertebrate Paleontology Collection at the Natural History Museum of Los Angeles County, as well as to Shannen Robson and Kayce Bell for their assistance with extant *Sigmodon* material in the Mammalogy Department (LACM). We also thank Brian Brown, Giar Ann, and Weiping Xie for facilitating the use of the Keyence microscope in the Entomology Department at LACM. Further acknowledgments go to Patricia Holroyd for reviewing material from the San Timoteo Formation at the University of California Museum of Paleontology, Berkeley, and to Laura Wilson for granting access to *Sigmodon minor* specimens housed at the Sternberg Museum in Kansas. Finally, we thank Ferhat Kaya and an anonymous reviewer for their valuable comments, which helped improve our final work.

## Author contributions

**Conceptualization:** Adolfo Pacheco-Castro, Oscar Carranza-Castañeda.

**Data curation:** Adolfo Pacheco-Castro.

**Formal analysis:** Adolfo Pacheco-Castro.

**Funding acquisition:** Xiaoming Wang.

**Investigation:** Adolfo Pacheco-Castro, Oscar Carranza-Castañeda.

**Methodology:** Adolfo Pacheco-Castro.

**Project administration:** Adolfo Pacheco-Castro, Oscar Carranza-Castañeda, Xiaoming Wang.

**Resources:** Xiaoming Wang.

**Supervision:** Oscar Carranza-Castañeda, Xiaoming Wang.

**Validation:** Adolfo Pacheco-Castro.

**Visualization:** Adolfo Pacheco-Castro.

**Writing – original draft:** Adolfo Pacheco-Castro.

**Writing – review & editing:** Adolfo Pacheco-Castro, Oscar Carranza-Castañeda, Xiaoming Wang.

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
