## [Decision Letter · Decision Letter 0]

18 Dec 2025

Dear Dr. Wang,

Thank you for submitting your manuscript to PLOS ONE. After careful consideration, we feel that it has merit but does not fully meet PLOS ONE’s publication criteria as it currently stands. Therefore, we invite you to submit a revised version of the manuscript that addresses the points raised during the review process.

We look forward to receiving your revised manuscript.

Kind regards,

Pablo Colunga-Salas

Academic Editor

PLOS One

Journal Requirements:

2. In your manuscript, please provide additional information regarding the specimens used in your study. Ensure that you have reported human remain specimen numbers and complete repository information, including museum name and geographic location.

For more information on PLOS One's requirements for paleontology and archeology research, see https://journals.plos.org/plosone/s/submission-guidelines#loc-paleontology-and-archaeology-research.

“Finally, we acknowledge the financial and logistical support provided by the Instituto de Geociencias through the DGAPA PAPIIT IN102425 project, Universidad Nacional Autónoma de México, and the National Science Foundation [EAR-1949742]. Additional funding for visiting localities and museums in the Great Plains was generously provided by the National Science Foundation [EAR-1655720].”

“National Science Foundation grant (EAR-1949742)”

“National Science Foundation grant (EAR-1949742)”

6. We note that Figures 1 and 5 in your submission contain map images which may be copyrighted. All PLOS content is published under the Creative Commons Attribution License (CC BY 4.0), which means that the manuscript, images, and Supporting Information files will be freely available online, and any third party is permitted to access, download, copy, distribute, and use these materials in any way, even commercially, with proper attribution. For these reasons, we cannot publish previously copyrighted maps or satellite images created using proprietary data, such as Google software (Google Maps, Street View, and Earth). For more information, see our copyright guidelines: http://journals.plos.org/plosone/s/licenses-and-copyright.

1. You may seek permission from the original copyright holder of Figure(s) [#] to publish the content specifically under the CC BY 4.0 license.

Additional Editor Comments:

Dear authors,

After a complex search for reviewers, I agree with two of them that the submitted work is of very high quality and will undoubtedly be a very important contribution to the study of sigmodontines. According to both reviewers, only a few minor details need to be addressed before accepting your manuscript for publication. I apologize for the delay, but I am certain your wait will be well worth it.

Reviewers' comments:

Reviewer's Responses to Questions

**Comments to the Author**

1. Is the manuscript technically sound, and do the data support the conclusions?

Reviewer #1: Yes

Reviewer #2: Yes

2. Has the statistical analysis been performed appropriately and rigorously?

Reviewer #1: Yes

Reviewer #2: No

3. Have the authors made all data underlying the findings in their manuscript fully available?

Reviewer #1: Yes

Reviewer #2: Yes

4. Is the manuscript presented in an intelligible fashion and written in standard English?

Reviewer #1: Yes

Reviewer #2: Yes

Reviewer #1: This manuscript presents the first discovery and documentation of Sigmodon minor from the early Blancan deposits of the San Miguel de Allende Basin (SMA), central Mexico, supported by radiometric age control ranging between 3.9 and 3.68 Ma. This represents a significant biochronological and paleobiogeographic contribution, extending the known range of S. minor well into southern North America and offering a new perspective on Blancan faunal exchanges.

The taxonomic identification is robust and well-supported through extensive morphological comparison with other Blancan and extant Sigmodon species. The conclusions regarding dispersal from the Great Plains and the paleoecological interpretation of open grassland-dominated habitats are consistent with the presented evidence and stratigraphic context.

The geological background and stratigraphic framework are well summarized, referencing both volcanic ash layers and associated index fossils (e.g., Nannippus peninsulatus, Glossotherium). These data provide solid age constraints and contextual support for the early Blancan assignment. The integration of radiometric and biostratigraphic data enhances the manuscript’s biochronological reliability.

The description of the maxillary and mandibular elements is detailed and precise, with clear specimen documentation (MPGJ series, UNAM collection). The authors provide adequate anatomical terminology and comparative analysis, which together justify the taxonomic identification of Sigmodon minor. The inclusion of both juvenile and adult morphotypes is particularly valuable for understanding intraspecific variation and ontogenetic trends.

However, a more quantitative component—such as plots or scatter diagrams of molar length vs. width—would significantly strengthen the analysis. This addition would illustrate variation across specimens and facilitate visual comparison with other Blancan populations (e.g., Meade Basin material). Even simple bivariate plots could reinforce the morphological conclusions already discussed in text form.

The paleobiogeographic synthesis is one of the manuscript’s strengths. The discussion effectively integrates Sigmodon minor occurrences across North America and identifies an asymmetrical dispersal pattern from the Great Plains toward both western and southern regions. Although largely qualitative, this discussion is well supported by the available data.

Typographical error at P25L425 (“Anterocone”) should be corrected.

This is a scientifically sound and significant paper, documenting an important southern occurrence of Sigmodon minor that enhances our understanding of Blancan rodent biogeography and early Pliocene faunal dynamics in North America. The study is technically solid, methodologically appropriate, and well contextualized within regional geology.

The manuscript will be ready for publication after these very minor refinements.

Reviewer #2: I have completed the review of the manuscript PONE-D-25-46840, entitled “First record of Sigmodon minor (Rodentia) in the early Blancan of central Mexico: asymmetrical dispersal from the Great Plains and paleoecology inferences”

This is a novel and valuable study, and I recommend it for publication in PLOS ONE. The authors present an important taxonomic, biogeographic, and paleoenvironmental analysis of cranio-dentary remains of the extinct cricetid rodent Sigmodon minor recovered from the early Blancan (Pliocene) of the San Miguel de Allende Basin, Guanajuato, Mexico. The morphological description of the fossil remains is of high quality. However, I have noted a few comments below, which I believe will help to improve the overall quality of the submission.

1. The authors propose that it is possible to distinguish between older and more recent populations of Sigmodon minor based on molar size, as this species appears to decrease in size over time or to be replaced by larger cotton rat species. However, no statistical analyses were conducted to evaluate size variation through time. I suggest applying a multivariate approach (e.g., principal component analysis).

2. The authors should provide details on the accumulation conditions. Specifically, how the microvertebrate remains were accumulated at the site (e.g., predation, in situ death, or reworking by post-depositional agents). Understanding the taphonomic history of the fossil assemblage, supported by robust evidence, is essential for interpreting paleoecological relationships and for refining paleoenvironmental reconstructions.

3. line 49: It should be clarified that the genus belongs to a sigmodontine cricetid, Sigmodon (Cricetidae, Sigmodontinae). Line 659: the genus name Sigmodon should be italicized.

4. Figure 1 should be improved: (1) the location of the fossil-bearing locality is too general. I suggest improving the map by increasing the zoom level and providing greater detail for the area where the record was found; (2) in the stratigraphic column on the left, italics should be removed from Sciuridae, as it is a suprageneric taxonomic category.

5. Figures 2 and 4: the main diagnostic characters (e.g., principal cusps, loph/ids, flex/ids, etc.) should be indicated in order to make the description easier to follow for non-specialist readers.

6. Figures 5 and 6: I suggest that the graphic quality and color of these figures be improved in order to better meet the standards of this journal.

Kind regards,

.

Reviewer #1: **Yes:**Ferhat KayaFerhat KayaFerhat KayaFerhat Kaya

Reviewer #2: No

---

## [Author Response · Author response to Decision Letter 1]

21 Feb 2026

Regarding the changes requested by the reviewers, the following modifications were made:

Reviewer 1

1. "However, a more quantitative component—such as plots or scatter diagrams of molar length vs. width—would significantly strengthen the analysis."

A paragraph addressing this issue was added to the Discussion section (Paleoecology and body size of Sigmodon minor, line 628), and a Supporting Information figure (S4 Fig) was included, entitled Mean m1 length of Sigmodon minor populations from the Rexroad (Kansas), GTO 19 (central Mexico), and Borchers (Kansas) localities.

2. "Typographical error at P25L425 (“Anterocone”) should be corrected."

The typographical error in “Anterocone” was corrected in line 426.

Reviewer 2

1. "The authors propose that it is possible to distinguish between older and more recent populations of Sigmodon minor based on molar size, as this species appears to decrease in size over time or to be replaced by larger cotton rat species. However, no statistical analyses were conducted to evaluate size variation through time."

A paragraph was added to the Methods section (line 170) describing that statistical analyses and figures were generated using custom scripts in R. A Welch ANOVA and Welch t-tests with Holm correction were conducted to compare groups from three localities: GTO 19 (Mexico), Borchers (Kansas), and Rexroad (Kansas). In addition, the paragraph originally located at line 618 in the Paleoecology and body size of Sigmodon minor section was replaced with new text incorporating the statistical results, allowing for a more robust discussion. A Supporting Information figure (S4 Fig), entitled Mean m1 length of Sigmodon minor populations from the Rexroad (Kansas), GTO 19 (central Mexico), and Borchers (Kansas) localities, was also included.

2. "The authors should provide details on the accumulation conditions. Specifically, how the microvertebrate remains were accumulated at the site (e.g., predation, in situ death, or reworking by post-depositional agents)."

The following description was added to the Geological Setting section (line 130):

“These localities are among the most significant in the region due to their high concentration of microvertebrate remains. Corrosion observed on some molar crowns suggests exposure to gastric acids, indicating that the accumulation at GTO 12 and GTO 19 likely resulted from predation by an unidentified predator. In contrast, at GTO 6, microvertebrate remains occur in stratigraphic concordance and in association with megafauna, suggesting accumulation through natural hydraulic sorting within an ancient fluvial system.”

3. "line 49: It should be clarified that the genus belongs to a sigmodontine cricetid, Sigmodon (Cricetidae, Sigmodontinae). Line 659: the genus name Sigmodon should be italicized."

The clarification was made in line 49 (Cricetidae, Sigmodontinae), and Sigmodon was italicized in line 688.

5. "Figures 2 and 4: the main diagnostic characters (e.g., principal cusps, loph/ids, flex/ids, etc.) should be indicated in order to make the description easier to follow for non-specialist readers."

We maintained the alphabetical order used to refer to these diagnostic structures according to their acronyms, as we consider this order to be appropriate and commonly used.

6. "Figures 5 and 6: I suggest that the graphic quality and color of these figures be improved in order to better meet the standards of this journal."

For Figures 5 and 6, we improved image quality and color. The most evident changes are observed in Figure 6, which was fully vectorized.

---

## [Editor Report · Decision Letter 1]

26 Mar 2026

First record of Sigmodon minor (Rodentia) in the early Blancan of central Mexico: asymmetrical dispersal from the Great Plains and paleoecology inferences

PONE-D-25-46840R1

Dear Dr. Wang,

We’re pleased to inform you that your manuscript has been judged scientifically suitable for publication and will be formally accepted for publication once it meets all outstanding technical requirements.

Kind regards,

Pablo Colunga-Salas

Academic Editor

PLOS One

Additional Editor Comments (optional):

Dear authors,

It is with great pleasure that we announce that, following peer review, your manuscript has been accepted. Congratulations on this contribution to the taxonomy and fossil knowledge of sigmodontines. We are confident that this work will be warmly received by the scientific community.
---

## [Editor Report · Acceptance letter]

PONE-D-25-46840R1

PLOS One

Dear Dr. Wang,

I'm pleased to inform you that your manuscript has been deemed suitable for publication in PLOS One. Congratulations! Your manuscript is now being handed over to our production team.

Kind regards,

on behalf of

Pablo Colunga-Salas

Academic Editor

PLOS One